# Clinical Use of a Commercial Artificial Intelligence-Based Software for Autocontouring in Radiation Therapy: Geometric Performance and Dosimetric Impact

**DOI:** 10.3390/cancers15245735

**Published:** 2023-12-07

**Authors:** S M Hasibul Hoque, Giovanni Pirrone, Fabio Matrone, Alessandra Donofrio, Giuseppe Fanetti, Angela Caroli, Rahnuma Shahrin Rista, Roberto Bortolus, Michele Avanzo, Annalisa Drigo, Paola Chiovati

**Affiliations:** 1Medical Physics Department, Centro di Riferimento Oncologico di Aviano (CRO) IRCCS, 33081 Aviano, Italy; smhasibul.hoque@cro.it (S.M.H.H.); giovanni.pirrone@cro.it (G.P.); rahnumashahrin.rista@cro.it (R.S.R.); mavanzo@cro.it (M.A.); adrigo@cro.it (A.D.); 2Radiation Oncology Department, Centro di Riferimento Oncologico di Aviano (CRO) IRCCS, 33081 Aviano, Italy; fabio.matrone@cro.it (F.M.); alessandra.donofrio@cro.it (A.D.); giuseppe.fanetti@cro.it (G.F.); angela.caroli@cro.it (A.C.); rbortolus@cro.it (R.B.)

**Keywords:** autocontouring, radiotherapy, artificial intelligence, time savings, dosimetry

## Abstract

**Simple Summary:**

Auto contouring driven by artificial intelligence can improve the workflow of radiotherapy by accelerating the contouring process. However, quality assurance of artificial intelligence-based tools is necessary for ensuring safety and efficacy in a clinical practice. In this study investigated the geometric accuracy of structural contours created by a commercial software for autocontouring based on artificial intelligence using well established metrics. In particular, the impact on the radiotherapy treatment plan quality from the adoption of artificial intelligence generated contours was investigated. Our results show that the combination of automatically generated contours and careful review by a clinical radiation oncologist results in time saving without affecting the quality of treatment plan. In conclusion, after quality checks that involve both geometric accuracy as well as dosimetric impact, contouring based on *AI* can be safely adopted in clinical practice.

**Abstract:**

Purpose: When autocontouring based on artificial intelligence (*AI*) is used in the radiotherapy (*RT*) workflow, the contours are reviewed and eventually adjusted by a radiation oncologist before an RT treatment plan is generated, with the purpose of improving dosimetry and reducing both interobserver variability and time for contouring. The purpose of this study was to evaluate the results of application of a commercial AI-based autocontouring for *RT*, assessing both geometric accuracies and the influence on optimized dose from automatically generated contours after review by human operator. Materials and Methods: A commercial autocontouring system was applied to a retrospective database of 40 patients, of which 20 were treated with radiotherapy for prostate cancer (PCa) and 20 for head and neck cancer (*HNC*). Contours resulting from *AI* were compared against *AI* contours reviewed by human operator and human-only contours using Dice similarity coefficient (*DSC*), Hausdorff distance (*HD*), and relative volume difference (*RVD*). Dosimetric indices such as *D_mean_*, *D*_0.03*cc*_, and normalized plan quality metrics were used to compare dose distributions from RT plans generated from structure sets contoured by humans assisted by *AI* against plans from manual contours. The reduction in contouring time obtained by using automated tools was also assessed. A Wilcoxon rank sum test was computed to assess the significance of differences. Interobserver variability of the comparison of manual vs. AI-assisted contours was also assessed among two radiation oncologists for PCa. Results: For PCa, AI-assisted segmentation showed good agreement with expert radiation oncologist structures with average *DSC* among patients ≥ 0.7 for all structures, and minimal radiation oncology adjustment of structures (*DSC* of adjusted versus *AI* structures ≥ 0.91). For *HNC*, results of comparison between manual and *AI* contouring varied considerably e.g., 0.77 for oral cavity and 0.11–0.13 for brachial plexus, but again, adjustment was generally minimal (*DSC* of adjusted against *AI* contours 0.97 for oral cavity, 0.92–0.93 for brachial plexus). The difference in dose for the target and organs at risk were not statistically significant between human and AI-assisted, with the only exceptions of D_0.03cc_ to the anal canal and *D_mean_* to the brachial plexus. The observed average differences in plan quality for PCa and *HNC* cases were 8% and 6.7%, respectively. The dose parameter changes due to interobserver variability in PCa were small, with the exception of the anal canal, where large dose variations were observed. The reduction in time required for contouring was 72% for PCa and 84% for *HNC*. Conclusions: When an autocontouring system is used in combination with human review, the time of the RT workflow is significantly reduced without affecting dose distribution and plan quality.

## 1. Introduction

Radiation therapy (RT) is considered as an alternative to surgery for early-stage cancer, whereas locally advanced cancer is mostly treated in conjunction with surgery and systemic radiation therapies according to patient’s age and comorbidities [1,2,3]. Improper delineation of the target volume and organs at risk (OARs) can affect the quality of dose distribution designed during planning of the RT treatment. As a consequence, inadequate target coverage or normal tissue sparing may occur, resulting in a reduced tumor control or an increased probability of side effects [4]. Traditionally, tumor volumes and OARs are manually contoured by radiation oncologists. This is a laborious procedure that is subject to both intra- and interobserver variability [5]. In this scenario, automatic contouring methods can minimize the clinical workload as well as improving reproducibility of RT. In the contouring workflow, the automatic contours depict a starting point, which is reviewed and, if necessary, manually edited before being sent to the treatment planning system.

Atlas-based contouring [6], statistical models of shape and appearance [7], artificial intelligence-based methods [8], and hybrid strategies are a few examples of the automated contouring techniques that have been introduced and developed with promising outcomes. The spread of artificial intelligence (*AI*) is impacting the workflow of RT treatment in several scenarios [9], and AI-based autocontouring software has been developed and made available to oncologists to optimize the contouring process [10]. A question that arises is whether the automated contours are of sufficient quality for clinical use, which can be answered only after effective validation, that is, evaluation of accuracy and reliability. The existing literature indicates that contour evaluation is performed mostly at the geometric level [11,12,13] using common geometric metrics, including moment-based methods, overlap metrics, and distance-based measures [14]. However, geometrical metrics alone do not necessarily reflect the actual clinical impact of the contour differences [11,12,13]. Treatment dosimetry, plan quality, and associated clinical decision-making processes are directly influenced by accuracy of contoured regions, and the impact of the geometric agreement into the dose domain and plan quality remains to be fully investigated [14,15,16,17].

### Research Objectives

With the capability of automatically providing contours that can be used to generate clinically acceptable plans, commercial tools for automated segmentation can reduce treatment planning time substantially. The objective of this study was to investigate the accuracy of structure contours generated by commercial autocontouring software. Also, we wanted to investigate the dose distributions of the treatment plans generated from autocontoured structure sets.

## 2. Materials and Methods

### 2.1. Patient Data

After approval from the institutional review board of Centro di Riferimento Oncologico (CRO), 40 patients treated at CRO Aviano from September 2017 to June 2022 were selected retrospectively for this study. A total of 20 had been treated for prostate cancer (PCa) and 20 for head and neck cancer (*HNC*). The PCa patients’ preparation before CT acquisition included full bladder and empty rectum. Patients with bilateral hip implants or rectal spacer were not included in the study. Patients with *HNC* cancer required no preparation but were immobilized with a thermoplastic mask. No contrast was administered for any patient before CT image acquisitions.

Patients CTs for planning of the treatment were acquired using a 90 cm wide bore Toshiba Aquilion 16 CT simulator with 5 mm slice thickness for PCa and 2 mm for *HNC*. Images were reconstructed using FC13 reconstruction algorithm having a 256 × 256 matrix.

### 2.2. Contouring Workflows

Target volume delineations are ruled by international guidelines and scientific associations recommendations. The contoured structures for PCa patients included the entire prostate and its capsule, which represent the clinical target volume (CTV) as well as the organs at risk. The planning target volume (*PTV*) was created by expanding CTV by 5 mm margin in all directions except 3 mm posterior. For *HNC*, organs at risk were contoured automatically, while the CTV was not automatically contoured.

Contoured structures, excluding the *PTV*, were generated for each patient using three methods, as follows:-Manual contouring (*C_man_*). Contours were delineated by a radiation oncologist with at least ten years of experience, also using semiautomated tools like flood fill and interpolation, within the integrated ARIA and Eclipse TPS systems (version: 16.1; Varian Medical Systems, Inc., NewYork, CA, USA) [18] and following the institutional guidelines [19,20,21]. These contours were assumed as the ground truth structures.-Fully automated contouring based on artificial intelligence (*C_AI_*). These were automatically created using a research version of Limbus Contour (version: 1.0.18; Limbus AI Inc., Regina, SK, Canada) [22] software. Limbus Contour (LC) employs organ-specific deep convolutional neural network models on the basis of a U-net architecture [23], which were trained on CT images from the Cancer Imaging Archive public database [24]. Following the creation of contours, LC applies a number of postprocessing techniques including as outlier removal, slice interpolation, z-plane cutoffs, and contour smoothing [23]. Contouring the structure set on a patient required up to 7 min on 3.2 GHz Intel Pentium CPU G3420; 4 GB RAM; 64-bit Operating System. The Limbus-generated RT structure sets were exported as DICOM files into the Varian Eclipse workstation for revision and validation.-automatically generated contours reviewed and eventually adjusted by radiation oncologist (*C_AI,adj_*). Expert ROs reviewed and, if necessary, modified the *C_AI_* using the Eclipse contouring application in accordance with institutional consensus guidelines for target volume and OAR contours. In order to keep ROs blinded to the original contours’ creation, only *C_AI_* contours were visible to them during revision time.

For better reproducibility, manual contouring was performed by one radiation oncologist for each treated site and interobserver variability was measured between two operators for PCa (Section 2.9).

### 2.3. Treatment Planning and Delivery

The radiotherapy plans that had been previously delivered to the patients were assumed as the reference for dose comparison. Structure sets for these treatments were manually contoured by radiation oncologists (ROs) using the institutional protocol for Radiotherapy Oncology Group. Treatment planning was performed using dose prescription and constraints for planning shown in Table 1 and Table 2.

PCa treatments were delivered using the volumetric modulated arc therapy (VMAT) technique with one or two 18 MV full coplanar arcs, 600 MU/min maximum dose rate, and a prescribed dose of 60 Gy in 20 fractions each of 3 Gy. *HNC* patients received intensity-modulated radiation therapy (IMRT) treatments with nine 6 MV photon beam fields, a maximum dose rate of 300 monitor units (MU) per minute, and a prescribed dose of 70.95 Gy in 33 fractions of 2.15 Gy each. These plans were generated using the Eclipse planning system (Varian Medical). Dose calculations were performed using the anisotropic analytical algorithm (AAA) with a grid resolution of 2.5  mm [25]. The treatment schedule consisted of 5 daily fractions per week. The treatments were administered using a Varian TrueBeam or Trilogy linear accelerator. A cone-beam computed tomography image was acquired at the beginning of each treatment session for image-guided RT [26]. For evaluation of dose difference due to autocontouring in the planning workflow, treatment plans were exactly same for the structure set *C_AI,adj_* following the same planning and optimization procedure as for the plans clinically used. Plans were exported for analysis in RT DICOM formats from the treatment planning system. DICOM files were transferred to a high-performance computer interface for analysis with homemade MATLAB scripts.

### 2.4. Qualitative Assessment of Automated Contouring

An experienced clinician assessed the *C_AI_* for each patient using a four-point Likert scale, shown in Table 3, to evaluate qualitatively the automated contouring process. As such a test aims at distinguishing between *AI* and human operator, this is sometimes referred to as the Turing test [27,28].

### 2.5. Geometric Evaluation

For target and OAR structures, comparisons between *C_man_* and *C_AI_* contours before and after the physician review (*C_AI,adj_*) were compared with these metrics described herein. For comparing AI- versus human-generated contours, we used different types of geometrical metrics that are based on distance between surfaces, size of overlapping volumes, and difference in size [29].

Dice similarity coefficient (*DSC*) provides a measure of the volumetric overlap of two contours of a structure with a score range from 0 (no overlay) to 1 (total overlay) [30]:(1)$$DSC\left(C_{man},C_{AI,adj}\right)=\frac{2\left|C_{man}\cap C_{AI,adj}\right|}{\left|C_{man}\right|+\left|C_{AI,adj}\right|}$$

Hausdorff distance (*HD*) is a bidirectional measure of distance between contour surfaces [30]. This metric calculates the distance to the closest point in both directions, from contour *C_man_* to contour *C_AI,adj_* and vice versa, to figure out the largest surface-to-surface separation between two contours.
(2)$$HD\left(C_{man},C_{AI,adj}\right)=\max\left[h\left(C_{man},C_{AI,adj}\right),h\left(C_{AI,adj},C_{man}\right)\right]$$
where *h (*$C_{man}$*,*$\;C_{AI,adj}$*)* represents the Euclidean distance between a and b voxels corresponding to the *C_man_* and *C_AI_*_/_*C_AI,adj_* contours, respectively, and the formula is:(3)$$h\left(C_{man},C_{AI,adj}\right)=max_{a\in C_{man}}min_{b\in C_{AI,adj}}||a-b||$$

Relative volume difference (*RVD*), also known as relative absolute volume difference, describes the size difference between the regions:(4)$$RVD=\left|\frac{\left|V_{AI,adj}\right|-\left|V_{man}\right|}{\left|V_{man}\right|}\right|$$
where *V_man_* and *V_AI,adj_* represents the absolute volume corresponding to the *C_man_* and *C_AI,adj_* contours, respectively.

### 2.6. Evaluation of Dose Differences

To assess the potential impact of *AI* on dosimetry, we calculated the difference in dose indexes among plans as:(5)$$∆D_{X}=\frac{{\left(D_{X}\right)}_{AI,adj}-{\left(D_{X}\right)}_{man}}{{\left(D_{X}\right)}_{man}}$$
where (*D_X_*)*_man_* and (*D_X_*)*_Ai,adj_* referred to dose parameters for *C_man_* and *C_AI,adj_* contours, respectively. And *X* represents the dose metrics such as *D_min_*, *D_mean_*, and *D*_0.03*cc*_.

Dose distribution to the organs-at-risk (OAR) doses were evaluated using *D_mean_* (mean dose) and the highest dose encompassing 0.03*cc*, *D*_0.03*cc*_ [31].

Homogeneity Index (*HI*) was utilized to evaluate the dose uniformity within the PTV. *HI* was assessed using the following formula [32]:(6)$$HI=\frac{\left(D_{2\%}-D_{98\%}\right)}{D_{50\%}}$$

Where *D*_2%_ (near maximum dose), *D*_98%_ (near minimum dose), and *D*_50%_ represent the minimum dose covering 2%, 98%, and 50% of the target volume, respectively. Formula for *HI* comparison between plans for *C_man_* and *C_AI,adj_* contours was-
(7)$$∆HI=\frac{\left|HI_{AI,adj}-HI_{man}\right|}{\left|HI_{man}\right|}$$
where *HI_man_* and *HI_AI,adj_* represent the *HI* for *C_man_* and *C_AI,adj_* contours, respectively.

Conformity Index (*CI*) was used to obtain a quantitative evaluation of the *PTV* coverage by the prescribed dose. *CI* was evaluated using the following equation [33]:(8)$$CI=\frac{{\left(TV_{PV}\right)}^{2}}{V_{PTV}\times V_{TV}}$$
where *V_TV_* indicates the volume that receives 95% of the prescribed dose, *V_PTV_* represents the *PTV* volume, and *TV_PV_* is the *PTV* volume inside the *V_TV_*. *CI* comparison between *C_man_* and *C_AI,adj_* contours plan was carried out by using the formula below:(9)$$∆CI=\frac{\left|CI_{AI,adj}-CI_{man}\right|}{\left|CI_{man}\right|}$$
where *CI_man_* and *CI_AI,adj_* indicates the *C_man_* and *C_AI,adj_* contours *CI*, respectively.

### 2.7. Normalized Plan Quality Metric

The plan quality metric (*PQM*) framework was designed to establish a standardized approach for assessing how well a particular treatment plan achieves specific dose volume objectives that serve as a hypothetical “virtual physician” [34]. A *PQM* scorecard is often created for every objective which assigns a score based on how effectively the objective is achieved by a particular plan. To enable meaningful comparisons across our study cases, we utilized the normalized *PQM* (*nPQM*) score, which divides the *PQM* score by the peak score achievable by the plan of a certain district (*PQM_max_*) and scales to the percentage. The formula used for the normalized plan quality metric was:(10)$$nPQM=\frac{PQM}{PQM_{max}}\times 100$$

The *PQM* scorecard to be used for analysis in this trial is shown in Table 4 and Table 5. To calculate the score for a particular objective, there are two different types of functions: threshold and linear score. The threshold score’s function awards no points if the objective is not achieved and the maximum number of points for the accomplished objective. The linear score’s function makes use of two thresholds. Maximum points are awarded if the plan satisfies the constraint’s “ideal threshold” and no points are assigned if it does not exceed the constraint’s “minimally acceptable threshold”. Using the value of the dose-volume statistic, linear interpolation between the two thresholds is used to calculate the number of scores awarded if the objective is between the two thresholds.

### 2.8. Evaluation of Contouring Time

The amount of time required for contouring was measured in order to estimate the increase in performance made possible with autocontouring. Since in clinical practice AI-generated contours need always to be reviewed and eventually modified by a radiation oncologist, we measured the reduction in contouring time from autocontouring as
(11)$$\mathrm{Reduction}\;\mathrm{in}\;\mathrm{time}\;=\frac{T(C_{man})-T(C_{AI,adj})}{T(C_{man})}$$

### 2.9. Interobserver Variability

The interobserver variability was assessed by comparing autogenerated structures reviewed and adjusted by two different operators. The geometric differences were calculated by assessing *DSC*, *HD*, and *RVD* among AI-assisted contours performed by the two operators:(12)$$DSC\left(1,2\right)=\frac{2\left|C_{AI,adj1}\cap C_{AI,adj2}\right|}{\left|C_{AI,adj1}\right|+\left|C_{AI,adj2}\right|}$$
where $C_{AI,adj1}$ is the contour generated by *AI* and adjusted by operator 1.

Dosimetric evaluation was performed using the same methods previously described. For instance, the interobserver variability for *D_min_* to an organ at risk was calculated as
(13)$$∆D_{\min}=\frac{\left|D_{\min,1}-D_{\min,2}\right|}{\left|D_{\min,2}\right|}$$
where $D_{\min,1}$ and $D_{\min,2}$ are the minimum doses to an organ at risk generated using *AI* and adjusted by operators 1 and 2, respectively.

### 2.10. Data Analysis

For geometrical and dosimetric evaluation, we developed an in-house script in MATLAB version R2021a (The MathWorks, Inc, Boston, MA, USA) [35] to compare structure sets and treatment plans for both the automatic and the manually edited contours as shown in Figure 1. Wilcoxon rank sum test was employed to perform dosimetric comparisons to determine if there were any significant differences between the individual OARs in each arm in terms of the *D_min_*, *D_mean_*, and *D*_0.03*cc*_ doses based on the reference dose distribution, and significant differences for the *PTV* doses in terms of *HI* and *CI* were assessed with the alpha (*α*) value 0.05 for 95% *CI.*

## 3. Results

### 3.1. Qualitative Assessment of Automated Contouring

Results of the quality assessment for the PCa and *HNC* contours are shown in Figure 2 and Figure 3, respectively.

### 3.2. Geometric Comparison

Table 6 and Table 7 show the differences among structures contoured with different modalities. The highest average *DSC* values were observed for the bladder and rectum, followed by the anal canal and prostate. The values of the average *HD* were 4.19 mm, 2.85 mm, and 1.08 mm for prostate, bladder, and rectum, respectively. The values of the *RVD* showed the same trend: 0.08, 0.02, 0.01 for prostate, bladder, and rectum, respectively.

Figure 4 shows *DSC*, *HD*, and *RVD* scores for PCa cases. For PCa, large variabilities in terms of *DSC* and *RVD* were observed for anal canal and both femur heads in comparison between *C_man_* and *C_AI,adj_* contouring. As for *HD* values, a wide range of values was reported for femur heads.

Table 8 provides a complete list of *DSC*, *HD*, and *RVD* values for *HNC* contours. The brain, mandible, parotids, and thyroid showed a high level of correlation with average *DSC* scores of 1.00, 0.98, 0.99, and 0.94, and average *HD* scores of 0.65 mm, 8.13 mm, 1.50 mm, and 9.58 mm, respectively, between the contours of before (*C_AI_*) and after physician review (*C_AI,adj_*). *RVD* values were generally close to 0.

Figure 5 shows the geometric evaluation results of *C_AI_* contour and *C_man_* contour both compared with *C_AI,adj_* contours. Autocontouring resulted in similar results for the brain, brainstem, mandible, and eyes (*DSC* > 0.83). For the brachial plexuses, parotids, cochlea and submandibular glands, there was a significant difference between the *C_man_* contour and *C_AI,adj_* contour, while other OARs had better performance in terms of *DSC*, *HD*, and *RVD*.

Table 9 summarizes the *DSC*, *HD*, and *RVD* values calculated between the *C_man_* and *C_AI,adj_* contours. The worst metrics were found in smaller structures such as lenses, while larger structures including the brain, mandible, eyes, and trachea showed a high level of correlation, with average *DSC* around 80% and lower *HD* and *RVD* values.

### 3.3. PTV Evaluation

Figure 6 shows the geometric and dosimetric comparison of the *C_man_* plan compared with *C_AI,adj_* for prostate *PTV* in terms of *DSC* and *RVD* scores and differences in *HI* and *CI*.

### 3.4. Dosimetric Comparison

Differences in *D_min_*, *D_mean_*, and *D*_0.03*cc*_ between manual and AI-assisted are shown in Figure 7.

The quantitative results of the dosimetric comparisons of plans with the *C_man_* plan compared with *C_AI,adj_* contours are summarized in Table 10. No significant dose differences were measured between manual and autocontour workflows, except the anal canal for PCa cases.

Differences in *D_mean_* to OARs for *HNC* between *C_man_* and *C_AI,adj_* contours are shown in Figure 8a, where the esophagus exhibited relatively large variations in ∆*D_mean_*. *D*_0.03*cc*_ to the eyes and cochleas had a difference of a maximum of 13% between the *C_man_* and *C_AI,adj_* plans (Figure 8b), while other OARs showed <10% differences, except for the constraint of contralateral brachial plexus and brainstem. Each circle symbol represents a value outside the standard deviation.

The dosimetric parameters of the *HNC* patients are listed in Table 11. The largest differences were seen in both brachial plexuses *D_min_* and *D_mean_*, with differences up to 82% and 35%, respectively, between the *C_man_* and *C_AI,adj_* pairs. The differences were relatively smaller for other OARs between the *C_man_* and *C_AI,adj_* contour plan pairs.

Differences between the achieved dosimetric parameters for PCa planning were not significant according to the Wilcoxon test, with the exception of *D*_0.03*cc*_ for the anal canal_._ Brachial plexuses showed significant differences in terms *D_mean_*. The statistical analysis results are shown in Table 12.

### 3.5. nPQM Comparison

*nPQM* revealed that all the plans optimized from *C_AI,adj_* were considered equivalent to *C_man_*, with only few plans deemed as inferior to the clinical plan but clinically acceptable. Table 13 summarizes the difference in plan quality for all study sites.

### 3.6. Time Savings

Table 14 reports the average times required for contouring over all test subjects with different methods in absolute and percentage units of time savings.

### 3.7. Interobserver Variability

The qualitative test results showed no significant difference between the two observers. Time saving percentages varied among ROs (from 64% to 72% and 16 to 19 min for PCa, respectively). Only 2% variation was observed in *nPQM*. A detail geometric differences are shown in Table 15.

As shown in Figure 9, the plans with *C_AI,adj_* resulted in anal canal coverage that largely differed from the manual contour plan. No significant geometric differences were found for *DSC* and *RVD* by comparison of both RO-reviewed contours with the *C_man_* contour. Table 16 tabulates the difference in dosimetric parameters for observer variability.

## 4. Discussion

Since automatic segmentation tools have become a more efficient alternative to expert manual segmentation, it is important that these applications undergo a thorough review, as the full responsibility of the use of *AI* falls to humans [36]. In particular, the medical physicists have the responsibility of a thorough quality assurance [37] and the radiation oncologist has clinical responsibility of the resulting contours. The purpose of this work was to explore the potential advantages of including an artificial intelligence-based autocontouring system in a clinical pathway in terms of time saving, contour generation accuracy, and radiotherapy plan quality obtained from such reviewed structure sets. The analysis was performed on a dataset of 40 cancer patients equally distributed for PCa and *HNC*.

As the majority of reported evaluation metrics in the literature are based on geometric metrics [38], and usually evaluate autocontouring without human intervention, we compared both the geometric and dosimetric plan quality performance of the autocontouring software (version: 1.0.18; Limbus AI Inc, Regina, SK, Canada), after physician validation and adjustment, against manual contours.

The first results of this work clearly indicate that with the aid of an AI-based autocontouring system, 72% and 84% of contouring time can be saved for PCa and *HNC* cases, respectively. More time saving is possible by implementing a fully integrated system that automatically detects the CT image by predefined protocol and contour structures, eliminating manual export/import function. Moreover, the geometric accuracy reached by Limbus *AI* showed a high compliance with the contours used in the clinical routine. The target and OARs of PCa patients were segmented to high geometric precision, with *DSC* between *C_man_* and *C_AI,adj_* ≥ 0.7. The anal canal contours had the largest differences, with an average value of *DSC* (0.70) as well as a 30% difference in volume between the *C_man_* and *C_AI,adj_* contours.

In comparison to AI-based *C_AI,adj_* contours, most of the structures of *HNC* cases, including the brain, mandible, eyes, and optic nerves, had a high degree of geometric correlation (*DSC* > 0.98, *HD* < 3.32 mm, and *RVD* near to 0). However, there were also structures with low *DSC*, such as the brachial plexus (*DSC* = 0.11–0.13), leading to a large variety of results, which is consistent with the previous literature [6,11]. The institutional recommendation to contour a larger larynx, for instance, may result in a poorer geometric correlation of this OAR. Moreover, for this study, the autocontouring software and the oncologist utilized only CT images without contrast enhancement for contouring and revision, while normally, ROs register MRI images to CT images for contouring the OARs.

In principle, the accuracy of contouring has a direct influence on plan optimization, and hence the assessment and decision-making process for treatment plans. As a result, the focus of this study was to determine whether *C_AI,adj_* contours could provide equivalent dosimetric findings to *C_man_* contours when examined using dosimetric parameters. The prostate *PTV* conformity index showed nearly no change in dosimetric analysis; however, there was a 22% difference in *HI*. Although this study did not contain target volume auto-segmentation, we exclusively examined prostate *PTV* for observation. The modest dosimetric variation in *PTV* might be attributed mostly to the expertise and different approach to planning by various medical physicists.

The greatest notable dose difference for PCa OARs’ dose-volume metrics was in the anal canal for the *C_man_* vs. *C_AI,adj_* contour plan, whilst other OARs maintained almost the same dose distribution. Femurs indicated slightly higher mean dose, which might be attributed to volume variance in the femur segmentation. In terms of *HNC* cases, both brachial plexuses showed a greater divergence in the mean dose for the *C_AI,adj_* contours as compared to the *C_man_* contours. Otherwise, no significant differences in dose-volume metrics were discovered for those plans. Dosimetric disparities between the *C_man_* and *C_AI,adj_* contour plans, on the other hand, were minimal for organs such as the cochlea, parotids, and submandibular glands. Only for the brachial plexus were mean dose differences statistically significant; otherwise, the Wilcoxon rank sum tests failed to identify a significant difference in the achieved dosimetric parameters between these plan pairs, implying that the *C_AI,adj_*-generated plans perform similarly to the *C_man_* contour in the dose optimization and evaluation process for *HNC* planning.

The complex interplay between structure geometry and dose distribution is reflected in the discrepancy between geometric and dosimetric performance. In addition to geometric accuracy, spatial dose distribution and steepness of dose gradients also affect dosimetry performance. Even if there is a significant difference in the dosimetric metrics between the *C_man_* and *C_AI,adj_* contours for a structure located far away from the high-dose zone, their absolute dosimetric values may be too small to have an impact on plan assessment and decision making. Furthermore, depending on whether it extracts point or volume-based dosimetry, each dosimetric parameter (i.e., maximum, mean, or volume-based parameter) has a distinct reliance and sensitivity to geometric change. For example, when the size of a structure varies in a high-dose gradient zone, the maximum dose may fluctuate more than the mean dose [4]. Overall, the complex interplay between structure geometry and dose distribution suggests that employing a commercial autosegmentation system that was not trained on local data necessitates further examination that includes both geometric and dosimetric analysis. This critical situation highlights the significance of adopting normalized plan quality metrics as a virtual physician that integrates both geometry and dosimetry assessment. The overall plan quality of PCa and *HNC* cases with the *C_AI,adj_* contour changed by 8.0% and 6.7%, respectively, when compared to the reference plan that was in a relatively acceptable range.

Interobserver variability analysis was conducted for PCa cases, where the geometric and dosimetric data acquired using each of the studied delineations by two ROs and the manual one was analyzed. Time savings and acceptance of AI-driven contours are approximately the same for both ROs. Except for the anal canal contour, there was a good correlation of geometric metrics (*DSC* > 0.92, *HD* < 3.74 mm, and *RVD* < 0.04) between two ROs. There was also a large dose variation (*D*_0.03*cc*_ was 12% and *D_mean_* was 23%) for the anal canal, despite the fact that the dose parameters for other OARs were identically matched between ROs. The overall normalized plan quality variation was 2% between ROs, whereas the difference between the *C_man_* and *C_AI,adj_* contour plan was 3.2%, suggesting that a standard starting point of contouring can reduce interobserver variability.

We considered manually delineated contours of CRO Aviano as the gold standard in this research. This is not to claim that manual delineation is “better” or “accurate” than AI-based delineation. Experts favored autosegmented contours over manual delineation for specific structures in our ongoing evaluation study. Manual delineation provides a clinically acceptable and recognized contour quality, implying some clinical expertise or local institution practices. As Limbus software (version: 1.0.18; Limbus AI Inc., Regina, SK, Canada) was trained using universal structure sets, software using local institutional datasets can lessen discrepancies because there are always some variances in practice between institutions.

This study has some limitations. Even if the selected cases for each district resulted in a homogeneous dataset, only a subgroup of the patients in this research were evaluated for dosimetry. Although it clearly highlighted the disparity between geometric metrics and dosimetry performance, further research including a wider pool of patient samples will be advantageous in characterizing the dosimetry performance of each unique structure. The contouring was carried out retrospectively using CT images without contrast enhancement and without the registration of MRI and/or PET images, which is now strongly suggested for the contouring of not only treatment volumes, but also OARs in some pathologies. For a more in-depth examination, research registering CT autocontouring with MRI and/or PET images might be a feasible option. To obtain a more complete scenario on how the performance of the Limbus autocontouring system affects the contouring procedure, a comparison with other similar software should be performed. Finally, the 5 mm CT slice thickness in the prostate patients, which is standard practice in our institution, is a relatively large value used in prostates [39]. A change in slice thickness from 5 to 3 mm has been shown to affect only the volume of the bladder significantly [40]. However, this should not affect the main conclusions of the present study, as the slice thickness was always consistent during the comparison among *AI* and humans in the prostate patients. Despite its limitations, this study offers a proof-of-concept methodology to investigate the impact of including in the RT workflow an autocontouring software.

## 5. Conclusions

In the contouring process, human assessment is required due to the lack of absolute dependability of automatic segmentation. Nonetheless, providing an approach that has the potential to speed up the contouring process in the vast majority of cases would be an improvement over present clinical practice.

The clinical acceptability and efficacy of the AI-driven approach are dependent on the structural segmentation for the site, and clinical criteria stringency, as demonstrated by the cancer sites. The varying performance of *C_AI,adj_* contours across structure sets suggests a different approach, in which automatic segmentation is used to generate a subset of contours where AI consistently performs well, and clinical effort is reserved for the complement subset, which may be more sensitive and subject to significantly larger error or variation.

Dose parameter analysis revealed that treatment plans optimized using AI-generated contours did not result in statistically significant differences when examined using normalized plan quality metrics. The results show that plans based on automatically generated contours do not overdose nearby OARs. However, no statistically significant link between geometric and dosimetric metrics was found. The outcomes from dosimetric analysis and interobserver variability suggest that AI-based autocontouring may help to establish a standard starting point for radiation therapy treatment.

## Figures and Tables

**Figure 1 cancers-15-05735-f001:**
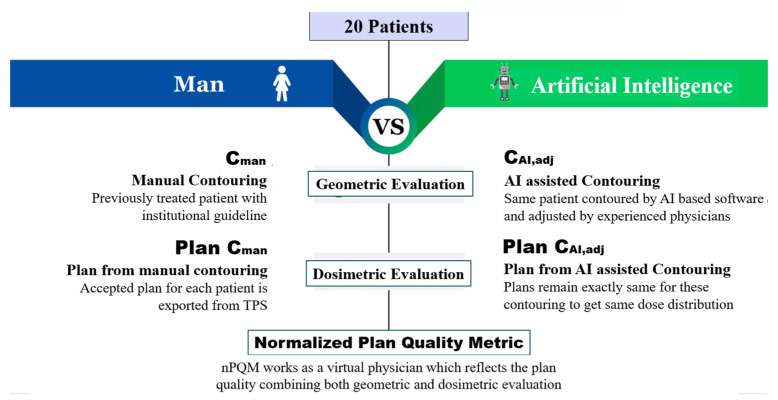
Comparison procedures for contouring evaluation.

**Figure 2 cancers-15-05735-f002:**
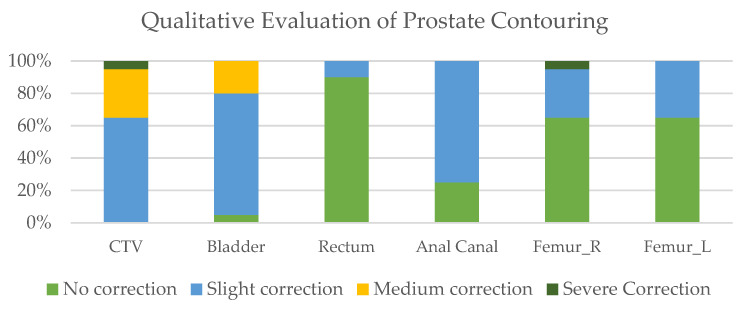
Evaluation of AI-based contouring for PCa.

**Figure 3 cancers-15-05735-f003:**
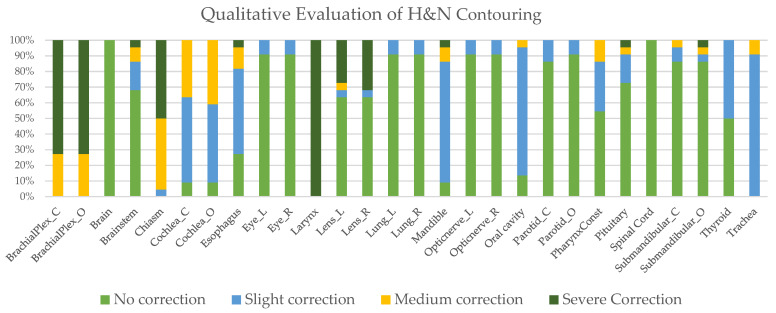
Physician assessment of AI-based contouring for *HNC*.

**Figure 4 cancers-15-05735-f004:**
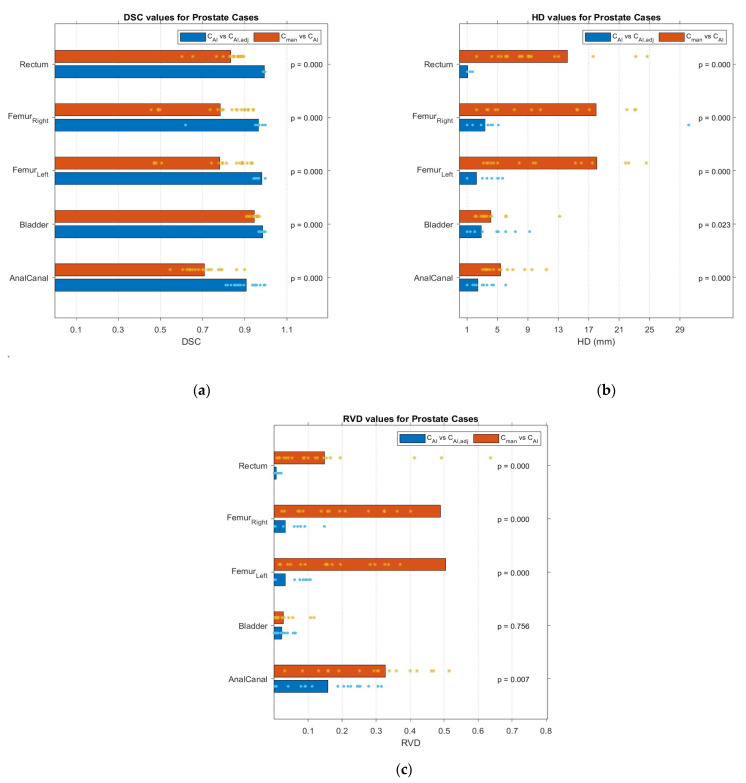
Geometric evaluation results: (**a**) *DSC*, (**b**) *HD* in mm, and (**c**) *RVD*, for *C_AI,adj_* contours in comparison with both *C_AI_* and *C_man_* contours of PCa cases. Each * represents a value.

**Figure 5 cancers-15-05735-f005:**
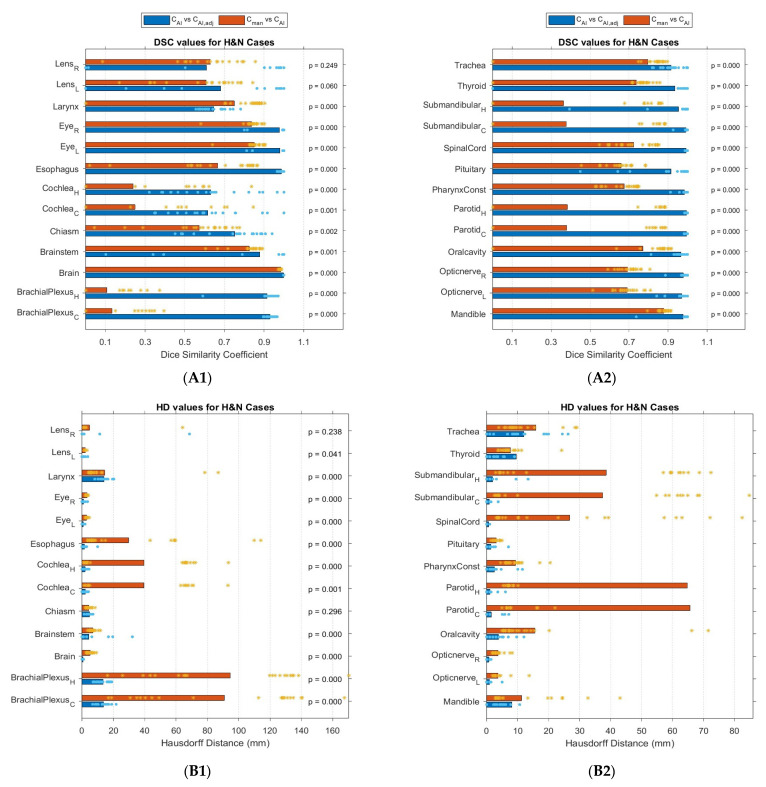
Geometric evaluation results: (**A1**,**A2**) *DSC*, (**B1**,**B2**) *HD* in mm, and (**C1**,**C2**) *RVD*, for *C_AI,adj_* contours in comparison with both *C_AI_* and *C_man_* contour of *HNC* cases. Each * represents a value.

**Figure 6 cancers-15-05735-f006:**
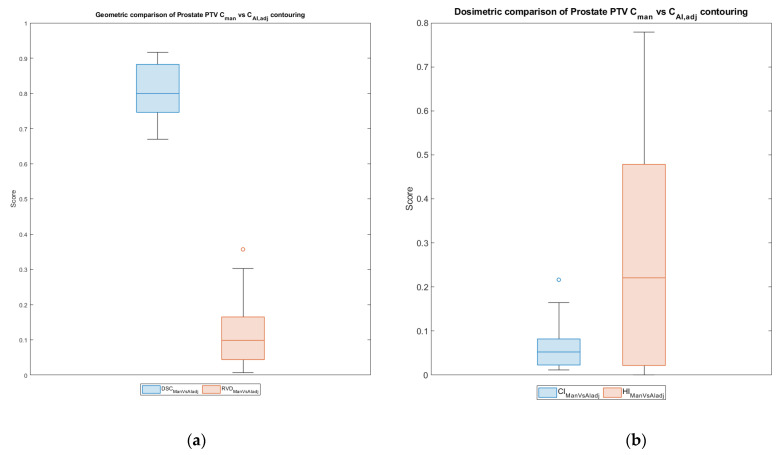
(**a**) Geometric evaluation by *DSC* and *RVD* and (**b**) dosimetric evaluation in terms of homogeneity index and conformity index of prostate PTV. Each circle symbol represents a value outside the standard deviation. Each circle symbol represents a value outside the standard deviation.

**Figure 7 cancers-15-05735-f007:**
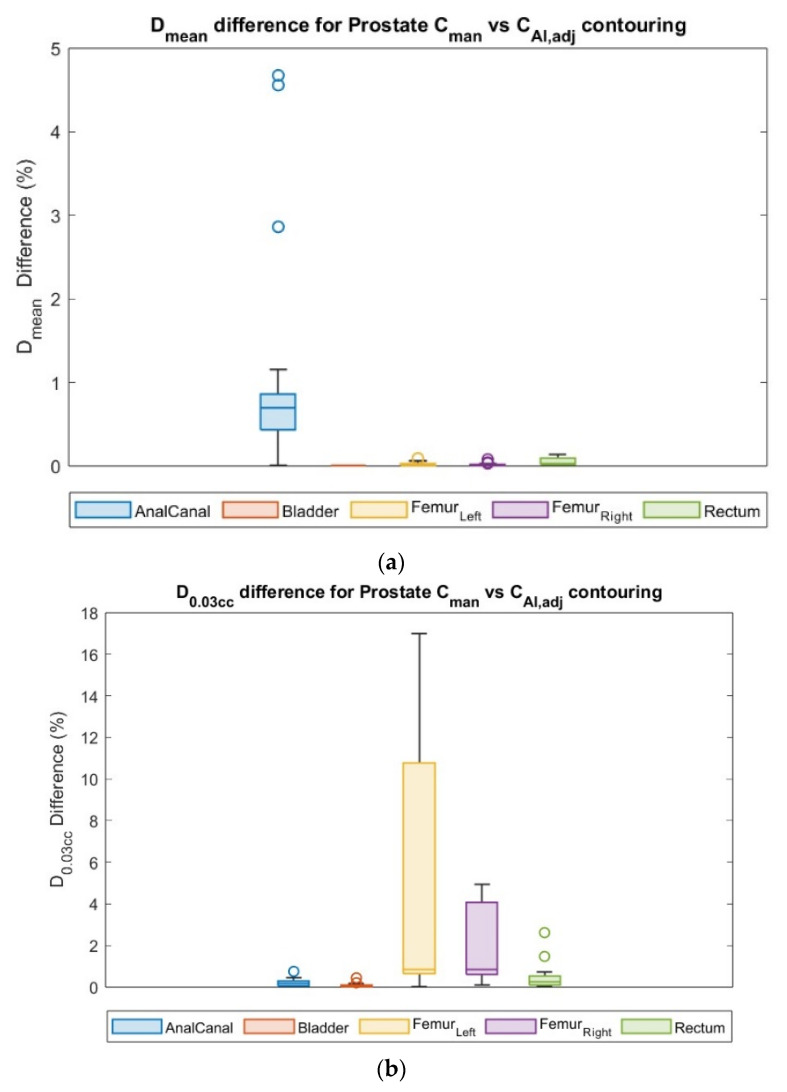
Dosimetric evaluation results: (**a**) relative difference in mean dose (*D_mean_*), and (**b**) relative difference in dose of 0.03*cc* volume (*D*_0.03*cc*_), for plan form *C_man_* contour in comparison with *C_AI,adj_* contours of PCa cases. Each circle symbol represents a value outside the standard deviation.

**Figure 8 cancers-15-05735-f008:**
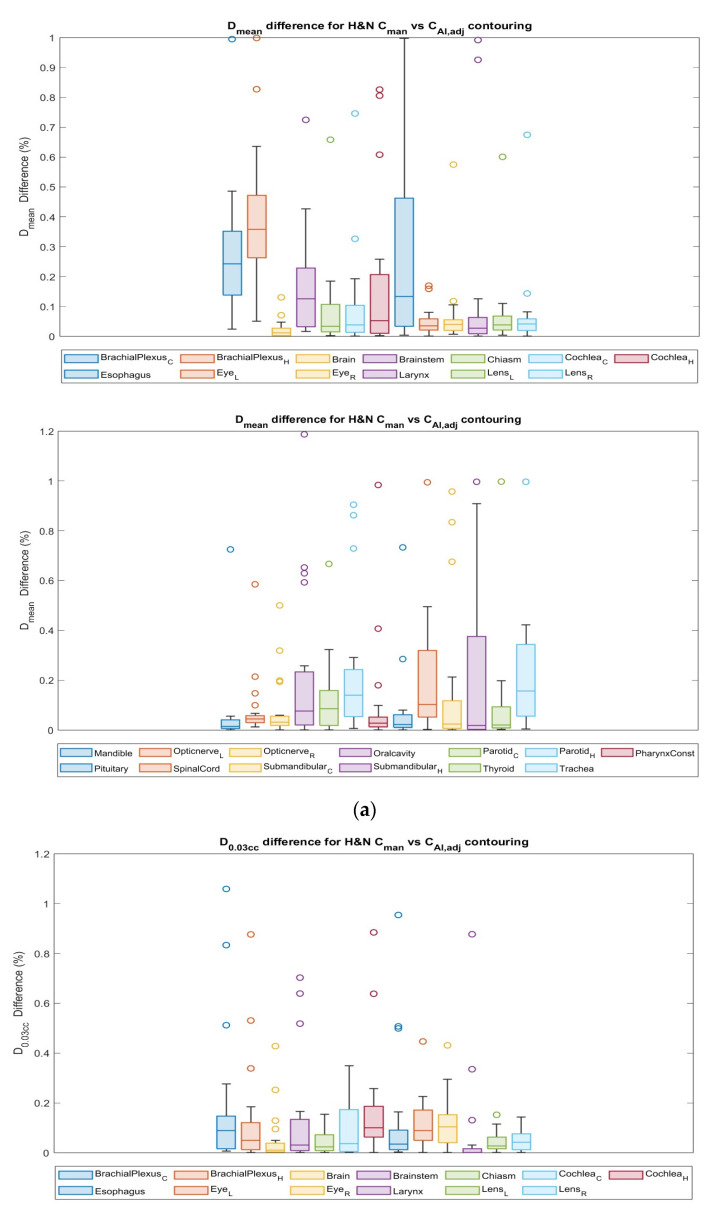
Dosimetric evaluation results: (**a**) relative difference in mean dose (*D_mean_*), and (**b**) relative difference in dose of 0.03*cc* volume (*D*_0.03*cc*_), for plan form *C_man_* contour in comparison with *C_AI,adj_* contours of *HNC* cases. Each circle symbol represents a value outside the standard deviation.

**Figure 9 cancers-15-05735-f009:**
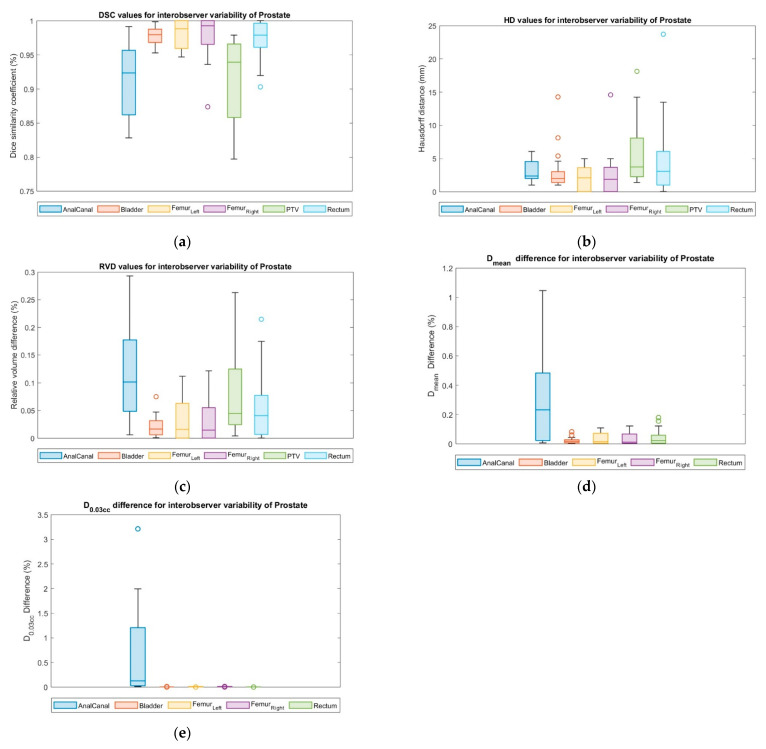
Geometric evaluation results: (**a**) *DSC*, (**b**) *HD* in mm, and (**c**) *RVD* and dosimetric evaluation results; (**d**) relative difference in mean dose (*D_mean_*) and (**e**) relative difference in dose of 0.03*cc* volume (*D*_0.03*cc*_) of interobserver variability. Each circle symbol represents a value outside the standard deviation.

**Table 1 cancers-15-05735-t001:** Dose constraints used for planning of PCa treatment.

Prostate (6000 cGy × 20 fx)	
PTV	D_min_ > 5700 cGy
	D_max_ < 6420 cGy
Rectum	V_4500 cGy_ < 15%
	V_2800 cGy_ < 35%
	V_900 cGy_ < 80%
Bladder	V_4500 cGy_ < 25%
	V_2800 cGy_ < 50%
Anal canal	V_4500 cGy_ < 15%
	V_2800 cGy_ < 35%
	V_900 cGy_ < 80%
Femoral head	V_3100 cGy_ < 1%

**Table 2 cancers-15-05735-t002:** Dose constraints used for planning of *HNC* treatment.

H&N (7095 cGy × 33 fx)	
PTV7095	D_min_ > 6953 cGy
	D_max_ < 7591 cGy
PTV6270	D_min_ > 5956 cGy
PTV5610	D_min_ > 5329 cGy
Brain	V_6000 cGy_ < 3%
Brainstem	D_max_ < 5000 cGy
Cochlea	D_mean_ < 4500 cGy
Spinal cord	D_max_ < 3600 cGy
PRV_Spinal cord	D_max_ < 4000 cGy
Oral cavity	D_mean_ < 3000 cGy
	V_3000 cGy_ < 73%
	V_4000 cGy_ < 20%
Ipsilateral parotid	D_mean_ < 2600 cGy
Contralateral parotid	D_mean_ < 2000 cGy
Ipsilateral submandibular gland	D_mean_ < 5000 cGy
Contralateral submandibular gland	D_mean_ < 3900 cGy
Mandible	V_5000 cGy_ < 31%
Arytenoid cartilage	V_5000 cGy_ < 50%
Constrictor muscle	V_5000 cGy_ < 70%
Constrictor muscle-PTV	D_mean_ < 5000 cGy
	V_5000 cGy_ < 31%
	V_5000 cGy_ < 31 cc
Thyroid	D_mean_ < 4500 cGy
	V_4000 cGy_ < 50%
	V_3000 cGy_ < 60%
Brachial plexus	V_6000 cGy_ < 0.1 cc
Esophagus	V_3500 cGy_ < 50%
	V_5000 cGy_ < 40%
	V_7000 cGy_ < 20%

**Table 3 cancers-15-05735-t003:** Scoring values for qualitative assessment of AI-generated contours.

Likert Scale for Each Patient		
Severe correction	: Require correction	→ Large and obvious errors
Medium correction	: Require correction	→ Minor errors that need a small amount of editing
Slight correction	: Accepted	→ Minor errors, but these are clinically not significant
No correction	: Accepted	→ Contour is very precise

**Table 4 cancers-15-05735-t004:** *PQM* of PCa treatment plans.

Structure	Constraint	Function	Thresholds	Max Score
PTV	D99%	Linear	>6000 cGY (100%)	5
			>5700 cGy (95%)	4
	D0.1 cc	Linear	<6300 cGy(105%)	5
			<6420 cGy(107%)	4
Rectum	V4500 cGy	Threshold	<15%	3
	V2800 cGy	Threshold	<35%	3
	V900 cGy	Threshold	<80%	3
Bladder	V4500 cGy	Threshold	<25%	3
	V2800 cGy	Threshold	<50%	3
Anal canal	V4500 cGy	Threshold	<15%	3
	V2800 cGy	Threshold	<35%	3
	V900 cGy	Threshold	<80%	3
Femur Right	V3500 cGy	Threshold	<1%	2
Femur Left	V3500 cGy	Threshold	<1%	2

**Table 5 cancers-15-05735-t005:** *PQM* of *HNC* treatment plans.

Structure	Constraint	Function	Thresholds	Max Score
BrachialPlexus_C	D_max_	Threshold	<6000 cGy	3
BrachialPlexus_O	D_max_	Threshold	<6270 cGy	3
Brain	V_6000 cGy_	Threshold	<3%	3
Brainstem	D_max_	Threshold	<5000 cGy	5
Chiasm	D_max_	Threshold	<5400 cGy	5
Cochlea_Contralateral	D_max_	Threshold	<1000 cGy	2
Cochlea_Ipsilateral	D_max_	Threshold	<3500 cGy	2
Esophagus	V_3500 cGy_	Threshold	<50%	1
	V_5000 cGy_	Threshold	<40%	1
	V_7000 cGy_	Threshold	<20%	1
Eye_L	D_max_	Threshold	<4500 cGy	5
Eye_R	D_max_	Threshold	<4500 cGy	5
Larynx	D_mean_	Threshold	<4000 cGy	2
	V_5000 cGy_	Threshold	<27%	2
Lens_L	D_max_	Threshold	<400 cGy	4
Lens_R	D_max_	Threshold	<400 cGy	4
Mandible	V_5000 cGy_	Threshold	<31%	3
OpticNerve_L	D_max_	Threshold	<5400 cGy	5
OpticNerve_R	D_max_	Threshold	<5400 cGy	5
Oral Cavity	D_mean_	Threshold	<3000 cGy	3
	V_3000 cGy_	Threshold	<73%	2
	V_4000 cGy_	Threshold	<20%	2
Parotid_Contralateral	D_mean_	Threshold	<2000 cGy	2
	D_median_	Threshold	<2000 cGy	4
Parotid_Ipsilatateral	D_mean_	Threshold	<2600 cGy	2
	D_median_	Threshold	<2600 cGy	4
PharynxConst	V_5000 cGy_	Threshold	<70%	2
Pituitary	D_max_	Threshold	<5000 cGy	5
SpinalCord	D_max_	Threshold	<4000 cGy	5
Submandibular_Co	D_mean_	Threshold	<3900 cGy	3
Submandibular_Ho	D_mean_	Threshold	<5000 cGy	3
Thyroid	D_mean_	Threshold	<4500 cGy	1
	V_4000 cGy_	Threshold	<50%	1
	V_3000 cGy_	Threshold	<60%	1

**Table 6 cancers-15-05735-t006:** Summary of *DSC*, *HD*, and *RVD* values measured before and after physician contours, (*C_AI_* vs. *C_AI,adj_*) for PCa cases.

	*DSC*		*HD* (mm)		*RVD*	
	Mean (SD)	Median (Range)	Mean (SD)	Median (Range)	Mean (SD)	Median (Range)
CTV	0.93 (0.07)	0.96 (0.68–0.99)	4.19 (3.31)	3.00 (1.0–11.09)	0.08 (0.11)	0.03 (0.0–0.46)
Rectum	0.99 (0.00)	0.99 (0.99–1.0)	1.08 (0.20)	1.00 (1.0–1.73)	0.01 (0.0)	0.01 (0.0–0.02)
Bladder	0.99 (0.01)	0.99 (0.97–1.0)	2.85 (2.54)	1.21 (1.0–9.22)	0.02 (0.02)	0.02 (0.0–0.06)
Anal Canal	0.91 (0.06)	0.89 (0.81–1.0)	2.44 (1.36)	2.00 (1.0–6.08)	0.16 (0.11)	0.20 (0.0–0.32)
Left Femur	0.98 (0.02)	1.00 (0.94–1.0)	2.23 (1.80)	1.00 (1.0–5.66)	0.03 (0.04)	0.00 (0.0–0.11)
Right Femur	0.97 (0.08)	1.00 (0.62–1.0)	3.34 (6.46)	1.00 (1.0–30.15)	0.03 (0.04)	0.00 (0.0–0.15)

**Table 7 cancers-15-05735-t007:** Summary of geometric difference metrics measured between *C_man_* and *C_AI,adj_* contours measured with different metrics for PCa cases.

	*DSC*		*HD* (mm)		*RVD*	
	Mean (SD)	Median (Range)	Mean (SD)	Median (Range)	Mean (SD)	Median (Range)
PTV	0.80 (0.07)	0.80 (0.67–0.91)	15.38 (9.20)	15.39 (3.16–37.7)	0.12 (0.10)	0.10 (0.01–0.35)
Rectum	0.83 (0.07)	0.85 (0.60–0.89)	14.20 (20.2)	8.62 (2.24–96.30)	0.15 (0.17)	0.09 (0.01–0.64)
Bladder	0.94 (0.01)	0.95 (0.90–0.97)	4.11 (2.49)	3.39 (2.0–13.19)	0.03 (0.03)	0.01 (0.0–0.12)
Anal Canal	0.70 (0.08)	0.70 (0.54–0.90)	5.43 (2.26)	4.47 (3.0–11.40)	0.33 (0.27)	0.30 (0.03–1.29)
Left Femur	0.78 (0.16)	0.83 (0.47–0.93)	18.06 (16.4)	12.64 (3.16–54.0)	0.50 (0.69)	0.18 (0.02–1.98)
Right Femur	0.78 (0.16)	0.84 (0.45–0.94)	17.98 (16.8)	13.01 (2.24–54.3)	0.49 (0.65)	0.20 (0.02–1.78)

**Table 8 cancers-15-05735-t008:** Summary of *DSC*, *HD*, and *RVD* values measured before and after physician contours (*C_AI_* vs. *C_AI,adj_*) for *HNC* cases.

	*DSC*		*HD* (mm)		*RVD*	
	Mean (SD)	Median (Range)	Mean (SD)	Median (Range)	Mean (SD)	Median (Range)
Contralateral Brachial Plexus	0.93 (0.02)	0.93 (0.90–0.97)	13.87 (3.80)	14.59 (6.78–21.8)	0.03 (0.02)	0.03 (0.00–0.08)
Ipsilateral Brachial Plexus	0.92 (0.08)	0.93 (0.59–0.97)	13.52 (4.12)	13.28 (7.07–19.1)	0.09 (0.18)	0.05 (0.01–0.82)
Brain	1.00 (0.00)	1.00 (1.00–1.00)	00.65 (0.49)	01.00 (0.00–1.00)	0.00 (0.00)	0.00 (0.00–0.00)
Brainstem	0.88 (0.27)	1.00 (0.10–1.00)	04.35 (8.54)	1.00 (0.00–32.16)	0.15 (0.31)	0.00 (0.00–0.95)
Chiasm	0.75 (0.15)	0.78 (0.45–0.94)	04.82 (1.40)	05.10 (1.41–7.00)	0.26 (0.12)	0.26 (0.08–0.44)
Contralateral Cochlea	0.62 (0.19)	0.60 (0.35–1.00)	02.21 (1.06)	02.24 (0.00–4.12)	0.51 (0.23)	0.57 (0.00–0.79)
Ipsilateral Cochlea	0.63 (0.20)	0.63 (0.32–1.00)	02.03 (0.96)	01.87 (0.00–4.58)	0.49 (0.24)	0.52 (0.00–0.81)
Esophagus	0.99 (0.01)	0.99 (0.97–1.00)	01.67 (2.12)	01.00 (0.00–10.0)	0.01 (0.02)	0.01 (0.00–0.07)
Left Eye	0.98 (0.05)	0.99 (0.81–1.00)	00.85 (0.59)	01.00 (0.00–2.00)	0.04 (0.12)	0.00 (0.00–0.46)
Right Eye	0.98 (0.06)	0.99 (0.80–1.00)	00.93 (0.82)	01.00 (0.00–3.61)	0.04 (0.13)	0.00 (0.00–0.50)
Larynx	0.65 (0.06)	0.63 (0.56–0.78)	14.11 (2.79)	13.83 (8.12–20.3)	0.49 (0.08)	0.51 (0.28–0.60)
Left Lens	0.68 (0.42)	0.93 (0.00–1.00)	01.00 (1.00)	01.00 (0.00–3.74)	0.34 (0.43)	0.06 (0.00–1.00)
Right Lens	0.61 (0.47)	0.95 (0.00–1.00)	5.96 (17.54)	01.00 (0.00–68.6)	0.39 (0.47)	0.05 (0.00–1.00)
Mandible	0.98 (0.06)	0.99 (0.74–1.00)	8.13 (19.10)	3.32 (0.00–88.21)	0.03 (0.09)	0.01 (0.00–0.41)
Left Optic Nerve	0.97 (0.04)	0.98 (0.84–1.00)	00.99 (1.06)	01.00 (0.00–5.00)	0.03 (0.06)	0.01 (0.00–0.22)
Right Optic Nerve	0.98 (0.03)	0.98 (0.89–1.00)	00.72 (0.49)	01.00 (0.00–1.41)	0.02 (0.04)	0.01 (0.00–0.20)
Oral Cavity	0.97 (0.04)	0.98 (0.81–1.00)	03.82 (3.07)	3.50 (0.00–12.04)	0.06 (0.07)	0.04 (0.00–0.31)
Contralateral Parotid	0.99 (0.00)	0.99 (0.98–1.00)	01.50 (2.02)	01.00 (0.00–7.14)	0.00 (0.01)	0.00 (0.00–0.02)
Ipsilateral Parotid	0.99 (0.00)	0.99 (0.98–1.00)	01.13 (1.43)	01.00 (0.00–6.08)	0.00 (0.00)	0.00 (0.00–0.02)
Pharynx Constrictor Muscle	0.98 (0.02)	0.99 (0.91–1.00)	02.57 (3.03)	1.21 (0.00–11.58)	0.02 (0.04)	0.01 (0.00–0.19)
Pituitary	0.92 (0.15)	0.97 (0.45–1.00)	01.33 (1.50)	01.00 (0.00–7.07)	0.10 (0.17)	0.01 (0.00–0.60)
Spinal Cord	0.99 (0.01)	0.99 (0.99–1.00)	00.65 (0.49)	01.00 (0.00–1.00)	0.00 (0.00)	0.00 (0.00–0.01)
Contralateral Submandibular	0.99 (0.02)	0.99 (0.93–1.00)	00.86 (0.84)	01.00 (0.00–3.74)	0.01 (0.02)	0.00 (0.00–0.09)
Ipsilateral Submandibular	0.95 (0.14)	0.99 (0.39–1.00)	01.91 (3.42)	1.00 (0.00–13.42)	0.05 (0.14)	0.00 (0.00–0.58)
Thyroid	0.94 (0.22)	0.99 (0.00–1.00)	9.58 (31.67)	01.62 (0.0–143.8)	0.11 (0.41)	0.01 (0.00–1.84)
Trachea	0.92 (0.05)	0.92 (0.82–1.00)	12.04 (7.25)	11.05 (0.0–26.29)	0.14 (0.08)	0.14 (0.00–0.31)

**Table 9 cancers-15-05735-t009:** Summary of geometric difference metrics measured between *C_man_* and *C_AI,adj_* contours measured with different metrics for *HNC* cases.

	*DSC*		*HD* (mm)		*RVD*	
	Mean (SD)	Median (Range)	Mean (SD)	Median (Range)	Mean (SD)	Median (Range)
Contralateral Brachial Plexus	0.13 (0.16)	0.00 (0.00–0.39)	90.86 (50.80)	119.9 (16.9–167)	0.97 (0.46)	0.87 (0.31–2.08)
Ipsilateral Brachial Plexus	0.11 (0.13)	0.00 (0.00–0.37)	94.65 (46.10)	120.6 (16.4–170)	1.05 (0.43)	1.07 (0.04–1.80)
Brain	0.98 (0.00)	0.99 (0.97–0.99)	05.20 (01.90)	04.79 (2.80–9.22)	0.01 (0.01)	00.01 (0.0–0.03)
Brainstem	0.83 (0.08)	0.85 (0.60–0.89)	06.99 (02.30)	6.44 (3.50–11.31)	0.17 (0.15)	0.12 (0.01–0.57)
Chiasm	0.57 (0.19)	0.63 (0.04–0.78)	04.50 (01.70)	04.12 (2.20–8.31)	0.20 (0.15)	00.20 (0.0–0.53)
Contralateral Cochlea	0.25 (0.31)	0.00 (0.00–0.85)	39.55 (34.70)	64.07 (1.4–93.09)	0.42 (0.27)	0.45 (0.02–0.87)
Ipsilateral Cochlea	0.24 (0.29)	0.00 (0.00–0.84)	39.66 (34.90)	64.55 (1.00–93.3)	0.48 (0.26)	0.51 (0.07–0.89)
Esophagus	0.67 (0.24)	0.79 (0.02–0.87)	29.76 (35.40)	8.33 (3.60–114.2)	5.62 (21.33)	0.22 (0.02–95.9)
Left Eye	0.85 (0.06)	0.85 (0.64–0.91)	02.97 (00.60)	03.00 (2.00–4.58)	0.24 (0.09)	0.25 (0.08–0.53)
Right Eye	0.84 (0.07)	0.85 (0.58–0.90)	03.17 (00.60)	03.08 (2.40–4.24)	0.26 (0.10)	0.25 (0.03–0.59)
Larynx	0.75 (0.26)	0.85 (0.00–0.90)	14.50 (23.50)	06.20 (4.0–86.98)	11.93 (36.3)	0.11 (0.03–121)
Lens_L	0.61 (0.19)	0.69 (0.17–0.84)	02.11 (00.60)	02.00 (1.40–3.16)	0.33 (0.32)	0.22 (0.03–1.41)
Lens_R	0.63 (0.17)	0.65 (0.08–0.86)	04.87 (14.00)	01.73 (1.0–64.33)	0.27 (0.23)	0.18 (0.02–0.96)
Mandible	0.88 (0.03)	0.88 (0.79–0.91)	11.29 (12.10)	04.12 (2.8–43.12)	0.17 (0.06)	0.17 (0.05–0.31)
Left Optic Nerve	0.69 (0.07)	0.69 (0.51–0.81)	03.68 (0270)	02.83 (2.0–13.78)	0.27 (0.15)	0.27 (0.02–0.54)
Right Optic Nerve	0.69 (0.06)	0.70 (0.59–0.81)	03.73 (01.90)	03.08 (1.70–8.31)	0.33 (0.12)	0.36 (0.05–0.51)
Oral Cavity	0.77 (0.27)	0.87 (0.00–0.92)	15.56 (18.70)	9.09 (5.0–71.510)	4.48 (15.36)	0.12 (0.02–66.7)
Contralateral Parotid	0.38 (0.43)	0.00 (0.00–0.89)	65.68 (52.30)	94.79 (4.9–147.8)	0.15 (0.08)	0.15 (0.03–0.31)
Ipsilateral Parotid	0.38 (0.44)	0.00 (0.00–0.89)	64.74 (54.10)	96.12 (5.4–148.1)	0.13 (0.08)	0.14 (0.01–0.35)
Pharynx Constrictor Muscle	0.67 (0.08)	0.71 (0.53–0.75)	09.34 (03.80)	9.06 (4.60–20.45)	0.19 (0.24)	00.14 (0.01–1.1)
Pituitary	0.66 (0.08)	0.67 (0.45–0.79)	03.08 (00.90)	03.00 (1.40–5.00)	0.29 (0.17)	0.33 (0.04–0.52)
Spinal Cord	0.72 (0.10)	0.71 (0.55–0.85)	26.80 (26.90)	11.63 (3.2–82.49)	0.27 (0.22)	0.23 (0.01–0.91)
Contralateral Submandibular	0.38 (0.43)	0.00 (0.00–0.89)	37.42 (31.20)	56.35 (2.5–84.73)	0.14 (0.10)	0.11 (0.02–0.47)
Ipsilateral Submandibular	0.36 (0.41)	0.00 (0.00–0.87)	38.61 (31.00)	58.19 (3.0–88.42)	0.19 (0.10)	0.20 (0.02–0.39)
Thyroid	0.74 (0.18)	0.77 (0.00–0.86)	07.68 (04.47)	06.63 (3.6–24.19)	0.14 (0.11)	0.11 (0.01–0.35)
Trachea	0.80 (0.19)	0.84 (0.00–0.90)	15.95 (18.90)	09.43 (4.0–90.63)	7.20 (31.34)	0.22 (0.0–140.3)

**Table 10 cancers-15-05735-t010:** Relative differences in *D_mean_* and *D*_0.03*cc*_ values measured between *C_man_* and *C_AI,adj_* contours for PCa cases.

	∆*D_mean_*		∆*D*_0.03*cc*_	
	Mean (SD)	Median (Range)	Mean (SD)	Median (Range)
Rectum	0.13 (0.17)	0.07 (0.0–0.73)	0.05 (0.05)	0.03 (0.0–0.13)
Bladder	0.03 (0.02)	0.02 (0.0–0.07)	0.00 (0.00)	0.00 (0.0–0.01)
Anal canal	0.49 (0.28)	0.48 (0.03–1.22)	1.11 (1.33)	0.69 (0.0–4.67)
Femur Left	0.21 (0.16)	0.19 (0.0–0.53)	0.02 (0.02)	0.01 (0.0–0.09)
Femur Right	0.21 (0.16)	0.21 (0.0–0.54)	0.01 (0.02)	0.01 (0.0–0.08)

**Table 11 cancers-15-05735-t011:** Summary of relative differences in *D_min_*, *D_mean_*, and *D*_0.03*cc*_ values for *C_man_* and *C_AI,adj_* contours for *HNC* cases.

	∆*D_mean_*		∆*D*_0.03*cc*_	
	Mean (SD)	Median (Range)	Mean (SD)	Median (Range)
Contralateral Brachial Plexus	0.25 (0.13)	0.24 (0.02–0.49)	0.12 (0.19)	0.08 (0.01–0.83)
Ipsilateral Brachial Plexus	0.35 (0.15)	0.35 (0.05–0.64)	0.07 (0.09)	0.02 (0.00–0.34)
Brain	0.02 (0.03)	0.01 (0.00–0.13)	0.06 (0.11)	0.02 (0.00–0.43)
Brainstem	0.14 (0.11)	0.13 (0.02–0.43)	0.14 (0.23)	0.04 (0.00–0.70)
Chiasm	0.05 (0.05)	0.03 (0.00–0.13)	0.05 (0.05)	0.04 (0.00–0.15)
Contralateral Cochlea	0.08 (0.09)	0.04 (0.00–0.33)	0.11 (0.11)	0.08 (0.00–0.35)
Ipsilateral Cochlea	0.11 (0.20)	0.04 (0.00–0.83)	0.16 (0.19)	0.10 (0.02–0.88)
Esophagus	0.18 (0.21)	0.08 (0.00–0.64)	0.09 (0.16)	0.03 (0.00–0.51)
Left Eye	0.04 (0.04)	0.03 (0.00–0.16)	0.13 (0.11)	0.09 (0.02–0.45)
Right Eye	0.04 (0.03)	0.03 (0.01–0.12)	0.13 (0.11)	0.10 (0.03–0.43)
Larynx	0.09 (0.23)	0.03 (0.00–0.93)	0.07 (0.23)	0.00 (0.00–0.88)
Left Lens	0.04 (0.03)	0.04 (0.00–0.11)	0.05 (0.04)	0.04 (0.00–0.15)
Right Lens	0.04 (0.03)	0.04 (0.00–0.14)	0.05 (0.04)	0.05 (0.00–0.14)
Mandible	0.02 (0.02)	0.01 (0.00–0.06)	0.01 (0.02)	0.00 (0.00–0.17)
Left Optic Nerve	0.05 (0.03)	0.04 (0.01–0.15)	0.07 (0.08)	0.06 (0.01–0.27)
Right Optic Nerve	0.05 (0.08)	0.03 (0.00–0.32)	0.07 (0.10)	0.04 (0.00–0.42)
Oral Cavity	0.11 (0.15)	0.04 (0.00–0.59)	0.05 (0.09)	0.02 (0.00–0.34)
Contralateral Parotid	0.10 (0.09)	0.10 (0.01–0.32)	0.11 (0.19)	0.05 (0.00–0.81)
Ipsilateral Parotid	0.15 (0.17)	0.09 (0.01–0.73)	0.10 (0.20)	0.02 (0.00–0.82)
Pharynx Constrictor Muscle	0.06 (0.10)	0.03 (0.00–0.41)	0.00 (0.01)	0.00 (0.00–0.05)
Pituitary	0.04 (0.07)	0.02 (0.00–0.28)	0.02 (0.02)	0.02 (0.00–0.07)
Spinal Cord	0.17 (0.17)	0.10 (0.00–0.50)	0.04 (0.06)	0.02 (0.00–0.26)
Contralateral Submandibular Gland	0.13 (0.24)	0.03 (0.00–0.83)	0.05 (0.07)	0.02 (0.00–0.26)
Ipsilateral Submandibular Gland	0.14 (0.20)	0.01 (0.00–0.64)	0.04 (0.06)	0.01 (0.00–0.22)
Thyroid	0.05 (0.06)	0.02 (0.00–0.20)	0.03 (0.05)	0.01 (0.00–0.18)
Trachea	0.17 (0.15)	0.13 (0.00–0.42)	0.05 (0.09)	0.02 (0.00–0.35)

**Table 12 cancers-15-05735-t012:** Statistical test results for *D_min_*, *D_mean_*, and *D*_0.03*cc*_ values measured between the plans generated from *C_man_* and *C_AI,adj_* contours of PCa and *HNC* cases.

Study Site	OAR	∆D_mean_	∆*D*_0.03*cc*_
Prostate	Rectum	0.64	0.35
	Bladder	0.90	0.37
	Anal Canal	0.11	0.04
	Left Femur	0.47	0.93
	Right Femur_R	0.23	0.82
Head and Neck	Contralateral Brachial Plexus	0.02	0.22
	Ipsilateral Brachial Plexus	0.00	0.41
	Brain	0.95	0.79
	Brainstem	0.92	0.84
	Chiasm	0.90	0.74
	Contralateral Cochlea	0.82	0.58
	Ipsilateral Cochlea	0.74	0.97
	Esophagus	0.34	0.66
	Left Eye	0.71	0.90
	Right Eye	0.86	1.00
	Larynx	0.51	0.47
	Left Lens	0.90	0.51
	Right Lens	0.95	0.52
	Mandible	0.92	0.94
	Left Optic Nerve	0.84	0.88
	Right Optic Nerve	0.80	0.79
	Oralcavity	0.52	0.39
	Parotid_C	0.86	0.42
	Parotid_H	0.30	0.44
	PharynxConst	0.99	0.92
	Pituitary	0.84	0.82
	SpinalCord	0.34	0.78
	Contralateral Submandibular gland	0.12	0.22
	Ipsilateral Submandibular gland	0.17	0.37
	Thyroid	0.95	0.66
	Trachea	0.92	1.00

**Table 13 cancers-15-05735-t013:** Relative difference in normalized plan quality metric between treatment plans with *C_man_* and *C_AI,adj_* contours.

Study Site	Mean (SD)	Median (Range)
Prostate	0.080 (0.097)	0.032 (0.00–0.276)
Head and Neck	0.067 (0.057)	0.054 (0.00–0.173)

**Table 14 cancers-15-05735-t014:** Time savings using AI-assisted autocontouring for study sites.

Study Site	*T_man_*	*T_AI,adj_*	Time Savings	Saved Time (%)
Prostate	23 min	6 min 25 s	16 min 35 s	72%
Head and Neck	2 h 30 min	23 min 35 s	2 h 6 min 25 s	84%

**Table 15 cancers-15-05735-t015:** Interobserver variability in terms of *DSC*, *HD* and *RVD* values measured between *C_AI,adj_* performed by two independent physicians for PCa cases.

	*DSC*		*HD* (mm)		*RVD*	
	Mean (SD)	Median (Range)	Mean (SD)	Median (Range)	Mean (SD)	Median (Range)
PTV	0.91 (0.06)	0.94 (0.80–0.98)	5.87 (4.70)	3.74 (1.41–18.14)	0.09 (0.09)	0.04 (0.0–0.26)
Rectum	0.97 (0.03)	0.98 (0.90–1.00)	4.76 (5.79)	3.08 (0.0–23.73)	0.06 (0.06)	0.04 (0.0–0.21)
Bladder	0.98 (0.01)	0.98 (0.95–1.00)	3.10 (3.17)	2.00 (1.0–14.28)	0.02 (0.02)	0.02 (0.0–0.08)
Anal Canal	0.91 (0.05)	0.92 (0.83–0.99)	3.11 (1.53)	2.34 (1.0–6.08)	0.12 (0.09)	0.10 (0.01–0.29)
Femur Left	0.98 (0.02)	0.99 (0.95–1.00)	2.02 (1.71)	02.12 (0.0–5.0)	0.03 (0.04)	0.02 (0.0–0.11)
Femur Right	0.98 (0.03)	0.99 (0.87–1.00)	2.53 (3.29)	1.87 (0.0–14.59)	0.03 (0.04)	0.02 (0.0–0.12)

**Table 16 cancers-15-05735-t016:** Summary of relative differences in *D_mean_* and *D*_0.03*cc*_ values measured between *C_AI,adj_* performed by two different radiation oncologists.

	∆*D_mean_*		∆*D*_0.03*cc*_	
	Mean (SD)	Median (Range)	Mean (SD)	Median (Range)
Rectum	0.04 (0.05)	0.02 (0.0–0.18)	0.00 (0.00)	0.00 (0.0–0.00)
Bladder	0.02 (0.02)	0.02 (0.0–0.08)	0.00 (0.00)	0.00 (0.0–0.01)
Anal Canal	0.28 (0.31)	0.23 (0.0–1.05)	0.65 (0.91)	0.12 (0.0–3.21)
Femur Left	0.04 (0.04)	0.01 (0.0–0.11)	0.00 (0.00)	0.00 (0.0–0.00)
Femur Right	0.04 (0.04)	0.01 (0.0–0.12)	0.00 (0.00)	0.00 (0.0–0.01)

## Data Availability

Data available on request due to privacy/ethical restrictions.

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
