# Peer review of "Clinical Use of a Commercial Artificial Intelligence-Based Software for Autocontouring in Radiation Therapy: Geometric Performance and Dosimetric Impact"

_cancers, 2023, doi:10.3390/cancers15245735_

Round 1

Reviewer 1 Report

Comments and Suggestions for Authors

In this article the authors present clinical use of a commercial artificial intelligence-based auto contouring software for radiation therapy of prostate cancer. They evaluate dosimetric, geometrical and time performance.

General comment:

The topic of this article is important to investigate; the article is well written and the limitations of the study are fairly presented.

Minor points:

Simple Summary is missing; the Abstract ‘should be a total of about 200 words maximum’.

Section 2.2: It is not written how many radiation oncologists (RO) performed manual contouring and reviewed and eventually adjusted automatically generated contours.

Lines 126, 149-150, 203-204, 249, 297-298, 308, 313 contain strange text.

Figure 1: The font size could be at least slightly increased (it is difficult to read).

Author Response

We thank both referees again and the editor for the positive feedback review of our revised manuscript.  We hope that we addressed all of the concerns adequately.

Reviewer 1#

Comment: In this article the authors present clinical use of a commercial artificial intelligence-based auto contouring software for radiation therapy of prostate cancer. They evaluate dosimetric, geometrical and time performance.

General comment:

The topic of this article is important to investigate; the article is well written and the limitations of the study are fairly presented.

Response: We thank the reviewer for the kind comment.

Comment: Minor points:

Simple Summary is missing; the Abstract ‘should be a total of about 200 words maximum’.

Section 2.2: It is not written how many radiation oncologists (RO) performed manual contouring and reviewed and eventually adjusted automatically generated contours.

Response: The contours were performed by one radiation oncologist for better repeatability and robustness. Then we separately assessed intra observer variability by comparing performance of AI assisted contouring  from two radiation oncologists. This was clarified as follows: “For ensuring better repeatability, the manual contours were performed by one radiation oncologist for each treated site and inter-observer variability was measured between two operators for pCA (section 2.9).”

Comment: Lines 126 (“as in table 1 and table 2”), 149-150 (Table 3), 203-204 (Table 4), 249 (Table 6 and Table 7), 297-298 (Table 12), 308 (Table 13), 313 (Table 14) contain strange text.

Response: We apologize but it seems that links to tables were broken during manuscript formatting. These were corrected as suggested.

Figure 1: The font size could be at least slightly increased (it is difficult to read).

Reviewer 2 Report

Comments and Suggestions for Authors

The author used an AI-based commercial auto contouring system to compare and verify dosimetric, geometrical and time performance in 20 prostate caner patients. Human only and contours resulting from AI were compared against AI contours reviewed by human operator using DSC, HD, and RVD values.

We have shown that using an automated contour system in combination with human review significantly reduces RT workflow time without compromising dose distribution and planning quality.

The purpose and research method of the study seem to be scientific and appropriate to the scope of the journal.

However, some issues must be modified to improve the quality of the paper.

 1. Line 125~126: “Treatment planning was performed using dose pre-125 prescription and constraints for planning shown in Error! There is an error “Reference source not found..” Please modify accordingly.

2. Line 249:” Error! Reference source not found.” There is an error.

3. In this study, 20 patients were enrolled, but we believe the number of patients is too small. Of course, limitations were mentioned in the research, but what do you think?

Comments on the Quality of English Language

none.

Author Response

We thank both referees again and the editor for the positive feedback review of our revised manuscript.  We hope that we addressed all of the concerns adequately.

Comment:The author used an AI-based commercial auto contouring system to compare and verify dosimetric, geometrical and time performance in 20 prostate caner patients. Human only and contours resulting from AI were compared against AI contours reviewed by human operator using DSC, HD, and RVD values.

We have shown that using an automated contour system in combination with human review significantly reduces RT workflow time without compromising dose distribution and planning quality.

The purpose and research method of the study seem to be scientific and appropriate to the scope of the journal.

Response: We thank the reviewer for the kind words.

Comment:However, some issues must be modified to improve the quality of the paper.

  1. Line 125~126: “Treatment planning was performed using dose pre-125 prescription and constraints for planning shown in Error! There is an error “Reference source not found..” Please modify accordingly.
  2. Line 249:” Error! Reference source not found.” There is an error.

Response: corrected references to Tables.

Comment: 3. In this study, 20 patients were enrolled, but we believe the number of patients is too small. Of course, limitations were mentioned in the research, but what do you think?

Response: We agree with these concerns regarding the limited dataset and scenarios. In order to address these concerns, 20 head and neck cancer patients were included in the analysis, thus bringing the total number of patients to 40 and increasing the diversity and difficulty of regions for contouring.